# Quantitative Computed Tomography Lung COVID Scores with Laboratory Markers: Utilization to Predict Rapid Progression and Monitor Longitudinal Changes in Patients with Coronavirus 2019 (COVID-19) Pneumonia

**DOI:** 10.3390/biomedicines12010120

**Published:** 2024-01-06

**Authors:** Da Hyun Kang, Grace Hyun J. Kim, Sa-Beom Park, Song-I Lee, Jeong Suk Koh, Matthew S. Brown, Fereidoun Abtin, Michael F. McNitt-Gray, Jonathan G. Goldin, Jeong Seok Lee

**Affiliations:** 1Department of Internal Medicine, College of Medicine, Chungnam National University, Daejeon 35015, Republic of Korea; ibelieveu113@cnuh.co.kr (D.H.K.); newcomet01@naver.com (S.-I.L.); goldjs2323@naver.com (J.S.K.); 2Department of Biostatistics, Fielding School of Public Health, University of California, Los Angeles, CA 90095, USA; gracekim@mednet.ucla.edu; 3Department of Radiological Sciences, David Geffen School of Medicine, University of California, Los Angeles, CA 90024, USA; mbrown@mednet.ucla.edu (M.S.B.); fabtin@mednet.ucla.edu (F.A.); mmcnittgray@mednet.ucla.edu (M.F.M.-G.); 4Center of Biohealth Convergence and Open Sharing System, Hongik University, Seoul 04401, Republic of Korea; sabeom@hongik.ac.kr; 5Graduate School of Medical Science and Engineering, Korea Advanced Institute of Science and Technology (KAIST), Daejeon 34141, Republic of Korea

**Keywords:** coronavirus disease 2019 (COVID-19), quantitative computed tomography (CT) score, rapid progression, prediction

## Abstract

Coronavirus disease 2019 (COVID-19), is an ongoing issue in certain populations, presenting rapidly worsening pneumonia and persistent symptoms. This study aimed to test the predictability of rapid progression using radiographic scores and laboratory markers and present longitudinal changes. This retrospective study included 218 COVID-19 pneumonia patients admitted at the Chungnam National University Hospital. Rapid progression was defined as respiratory failure requiring mechanical ventilation within one week of hospitalization. Quantitative COVID (QCOVID) scores were derived from high-resolution computed tomography (CT) analyses: (1) ground glass opacity (QGGO), (2) mixed diseases (QMD), and (3) consolidation (QCON), and the sum, quantitative total lung diseases (QTLD). Laboratory data, including inflammatory markers, were obtained from electronic medical records. Rapid progression was observed in 9.6% of patients. All QCOVID scores predicted rapid progression, with QMD showing the best predictability (AUC = 0.813). In multivariate analyses, the QMD score and interleukin(IL)-6 level were important predictors for rapid progression (AUC = 0.864). With >2 months follow-up CT, remained lung lesions were observed in 21 subjects, even after several weeks of negative reverse transcription polymerase chain reaction test. AI-driven quantitative CT scores in conjugation with laboratory markers can be useful in predicting the rapid progression and monitoring of COVID-19.

## 1. Introduction

The coronavirus disease 2019 (COVID-19) outbreak, caused by severe acute respiratory syndrome coronavirus 2 (SARS-CoV-2), has rapidly spread to become a global pandemic [1]. In May 2023, the World Health Organization (WHO) declared COVID-19 as an endemic and announced the end of the global public health emergency. Despite this declaration, some patients with COVID-19 still undergo rapid deterioration to severe disease with or without acute respiratory distress syndrome (ARDS) within 1–2 weeks from the onset of symptoms [2,3]. Many patients with severe diseases require the intensive care unit (ICU) care; therefore, predicting rapid progression in patients with confirmed COVID-19 is important. In a situation with a limited number of beds and an increase in the number of patients with COVID-19, it is necessary to recognize patients who are likely to progress to severe disease early and hospitalize and monitor them rather than classify all patients according to baseline characteristics, such as age and underlying disease.

In some papers published in 2020, the prevalence of severe COVID-19 was reported to be 15.7–26.1% in hospitalized patients in China [4,5,6]. These patients exhibited abnormal laboratory findings, including leukocytosis, neutrophilia, elevated d-dimer, and elevated procalcitonin, as well as chest computed tomography (CT) findings showing bilateral distribution of patchy shadows or ground glass opacity [5]. Respiratory failure due to severe COVID-19 pneumonia is associated with hyperinflammation and increased levels of cytokines, chemokines, and inflammatory mediators [7]. Since early identification of the risk of progression to severe disease is crucial in preventing respiratory failure and lowering mortality in patients with COVID-19 pneumonia, several studies have investigated whether severity can be predicted using various inflammatory markers such as C-reactive protein (CRP), D-dimer, interleukin-6 (IL-6), ferritin, Lactate dehydrogenase (LDH), neutrophil count, and lymphocyte count [8,9,10], and a scoring system was proposed based on this [11]. Elevated D-dimer, CRP, LDH, and high-sensitivity cardiac troponin I levels have been reported to be significantly associated with worse outcomes and alterations in white blood cell (WBC) count, and liver enzyme changes have been reported to be associated with severity [8]. However, the cut-off for each laboratory marker predicting severity has been reported differently in each study because different methods were used to identify those thresholds. Even for the widely used inflammatory marker, CRP, the cut-off ranged from 1.43 mg/dL to 80.5 mg/dL, indicating significant variability [5,9,10,12].

Since the COVID-19 pandemic, studies have been conducted to examine the relationship between various CT findings and clinical outcomes of patients, and there has been an increased interest in the prognostic value of chest CT at the time of COVID-19 diagnosis. Various studies have shown that several CT findings (scattered bilateral distribution of lesions, a higher number of involved lobes, the coexistence of diffuse GGO and consolidations, absence of mixed and reticular patterns, crazy paving, bronchus distortion, etc.) are associated with a poor prognosis in patients admitted to the ICU [13,14,15,16]. In addition, studies on pathophysiology including radiological–pathological signatures, and correlations between quantitative CT metrics and lung function in COVID-19 patients have been reported [17,18,19,20]. However, it is challenging to standardize CT findings as a predictor because each study’s description of the CT findings differs. Several other studies have evaluated the prognostic value of baseline chest CT using semi-quantitative methods (assigning specific scores according to the percentage of involved parenchyma) [21,22]. Recently, chest CT image analysis using multiple artificial intelligence (AI) models has stratified patient risk [23]; however, it is necessary to investigate the clinical utility of chest CT as a predictive tool for stratifying patients. Over the course of the prolonged COVID-19 pandemic, various studies have reported on the longitudinal changes in chest CT scans of patients who have experienced COVID-19 infection [24,25]. However, a limited number of studies have investigated these changes using serial clinical parameters and quantitative CT scores.

The novelty of this study is to explore the prognostic value of quantitative CT (QCT) lung COVID scores, along with laboratory inflammation markers including WBC, neutrophil count, lymphocyte count, CRP, procalcitonin, and IL-6, for predicting rapid progression within one week of hospitalization. Furthermore, we investigated the long-term changes and COVID-19 pneumonia based on the pairing of longitudinal CT imaging and laboratory data. This can infer the changes in CT with only the changes in accessible laboratory data.

## 2. Materials and Methods

### 2.1. Patients

This study included patients diagnosed with COVID-19 pneumonia and admitted to Chungnam National University Hospital (Daejeon, Republic of Korea) between February and September 2020, and their medical records were analyzed retrospectively. During this period, there were no variants of COVID-19 in South Korea [26]. The diagnosis of COVID-19 was established based on the positive results of the real-time reverse transcription polymerase chain reaction (RT-PCR) assay for SARS-CoV-2 in nasal and pharyngeal swab specimens.

This study was conducted in accordance with the Declaration of Helsinki and Good Clinical Practice guidelines and was approved by the institutional review board of our institution on 16 September 2020 (IRB No. 2020-09-041 at CNUH). Clinical and Digital Imaging and Communications in Medicine (DICOM) radiological data were anonymized according to the standards of care.

### 2.2. Clinical Data Collection

Patient demographic information (age, sex), body mass index (BMI), comorbidities, clinical symptoms and signs, and laboratory data (white blood cell count, platelet count, neutrophil count, lymphocyte count, C-reactive protein (CRP), procalcitonin, and interleukin-6 (IL-6) were obtained with data collection forms from electronic medical records.

Rapid progression was defined as a case in which the patient developed respiratory failure requiring intubation and mechanical ventilation within one week of hospitalization. Patients were classified into those with rapid progression and those without such a complication.

### 2.3. Quantitative High-Resolution Computed Tomography Imaging Analyses

Standardized non-contrast volumetric chest high-resolution computed tomography (HRCT) was performed with 3-mm slice thickness with equipment from 4 manufacturers (Siemens from München, Germany, GE from Boston, USA, Philips from Amsterdam, Netherlands, Toshiba from Tokyo, Japan). Baseline (the initial scan on hospitalization) and longitudinal follow-up HRCT scans were obtained, where the median (±IQR) follow-up duration from baseline was 8 (±5) days, 20 (±26) days, 51 (±21) days, and 60 (±36) days for 44% (97/218), 24% (53/218), 13% (29/218), and 5% (10/218), respectively, of the present study cohort. All scans were anonymized and digitally transferred for quantitative analyses. Quantitative HRCT analyses were performed by the Center for Computer Vision and Imaging Biomarkers (CVIB) at the Department of Radiological Science at the University of California Los Angeles (UCLA) (Computer Aided Analyses for CT Images, IRB No. 11-0000126 at UCLA). We developed the quantitative COVID scores from the HRCT images (Figure 1) using semi-supervised learning from the previous interstitial lung disease model. The adaption steps were to: (1) run the technique developed for interstitial lung disease [27,28] to infectious diseases [29], (2) identify misclassified regions of interest (ROI) in the infectious disease cohort, (3) explore the characteristics of misclassified ROIs, (4) make a simple adjustment for a new class, and (5) adapt this model by adding a new class of consolidation and expand GGO patterns. The final model to calculate the quantitative COVID scores was reviewed and confirmed the visualization of the classification result using an independent COVID-19 cohort at UCLA by thoracic radiologists (Jonathan G. Goldin and Fereidoun Abtin) [30]. High-throughput computation for COVID was performed using the pipeline of automated imaging import, labeling of the inspirational scans, and deep learning-based lobar segmentation [31]. Briefly, four steps of high-throughput are: (1) HRCT images are segmented by convolutional neural networks (CNN) after automatic labeling inspirational series using DICOM information [31]; (2) denoise HRCT images in order to normalize the different inherent noise [32]; (3) run adapted classifier model for COVID-19 using the denoised radiomic features and support vector machine [27]; (4) generate a score in ratio where the numerator is the number of classified voxels for a subtype and the denominator is the total number of voxels. For example, QGGO is a ratio where the numerator is the count of voxels that were classified as GGO patterns. In HRCT images from the COVID population, ground glass opacities usually represent acute inflammatory processes; mixed diseases represent heterogeneous opacity with areas of reticulation, with and without architectural distortion; and consolidation is frequently associated with pulmonary infection [33]. Quantitative COVID-19 Image (QCOVID) scores were composed of four distinct radiological patterns on HRCT: (1) quantitative ground-glass opacity (QGGO, yellow and cyan dots), (2) mixed diseases (QMD, red and blue dots), (3) consolidation (QCON, mint dots), and (4) normal lung (QNL). The sum of the three abnormal lung tissue scores was named the quantitative total lung disease (QTLD).

### 2.4. Statistical Analysis

Summary statistics were reported for the baseline characteristics. The mean and standard deviation (SD) and the median and interquartile ranges (IQR) were reported for the continuous dataset. *T*-tests were used for variables following the normal distribution, and Wilcoxon signed rank-sum tests were used for the variables that did not follow the normal distribution. Categorical variables were reported as numbers and percentages, and Fisher’s exact test was used for comparisons between the rapidly progressive and non-rapid progressor groups. Non-parametric Spearman rank correlations were used to associate QCOVID scores and laboratory markers. Logistic regression models were used to identify the important baseline factors of rapidly and non-rapid progressive populations using a forward variable selection with 0.15 as a cutoff threshold. ROC analyses were used to evaluate the performance in predicting rapid progression, and the area under the curve (AUC) was calculated. The optimal cut-off value of the QMD score was determined as the point at which the Youden index was maximized by the ROC curve. The upper limit of the normal range for inflammatory markers was set as the cut-off. Multivariable linear regression was used to test the important factors of clinical outcomes, QCT, and laboratory measurements. A mixed-effects linear model with a random intercept was applied for longitudinal changes from baseline. Stata software (version 17.0, College Station, TX, USA) was used to analyze the results.

## 3. Results

### 3.1. Patient Characteristics

A total of 225 patients with COVID-19 pneumonia were identified according to the inclusion criteria, and seven were excluded for the following reasons: (1) CT scores were not available due to slice thickness >3 mm due to the lack of robustness in radiomic features in thick slice [34] or no available DICOM images (*n* = 5); (2) different baseline scan date (*n* = 2). In total, 218 patients were included in our study (Appendix A). Baseline clinical and CT characteristics are shown in Table 1. The average length of hospital stay in all patients was 14.7 days; 34 patients (15.6%) required O_2_ demand, of which 21 patients (9.63%) received mechanical ventilation, and extracorporeal membrane oxygenation (ECMO) was applied to 5 patients (2.29%). The mortality rate was 1.83% (4/218 patients).

### 3.2. Rapid Progression Patients

Rapid progression was seen in 9.6% (21/218 patients); all patients received mechanical ventilation, and five received ECMO support. The age, WBC count, neutrophil count, and NLR were significantly higher in the rapidly progressive group, while the lymphocyte count was lower than in the non-rapid progressor group. The levels of CRP, procalcitonin, and IL-6 were significantly higher in the rapidly progressive group than in the non-rapid progressors, and QGGO, QMD, QCON, and QTLD scores were all significantly higher (Table 1). There were no significant differences in comorbidities between the two groups.

### 3.3. Association between Quantitative CT Lung COVID Score and Laboratory Findings

The QGGO, QMD, QCON, and QTLD scores showed a positive correlation with neutrophil count, and QMD, QCON, and QTLD scores showed a negative correlation with lymphocyte count. All four QCT COVID scores showed a significant positive correlation with NLR, CRP, and IL-6 levels (Table 2). Among those, the correlation coefficients of QMD score, CRP, and IL-6 were relatively higher (rho = 0.5669 and 0.4908, respectively) than the correlation coefficients of other parameters. Figure 2 displays the associations between QCT COVID scores (QMD and QTLD) and laboratory measurements of neutrophils, lymphocytes, NLR, CRP, IL-6, and PaO_2_ at baseline. Both the QMD score and QTLD score showed positive correlations with neutrophil count, neutrophil percentage, NLR, CRP, and IL-6. In contrast, both scores exhibited negative correlation with lymphocyte count, lymphocyte percentage, and PaO_2_. There are a few outliers shown in Figure 2. These outliers indicate that some patients with severe infections have extremely high levels of CRP and neutrophil count.

### 3.4. Prediction of Rapid Progression

In univariate analysis of quantitative CT score to predict rapid progression, all QCOVID scores predicted rapid progression, with the QMD score having the best predictive power (AUC 0.813, 95% confidence interval [CI] 0.679–0.947, *p* < 0.001) (Table 3). In the multivariate analysis, the QMD score and IL-6 level were important factors in predicting rapid progression (AUC = 0.864, 95% CI 0.775–0.953). Patients with a high QMD score (≥10%) were likely to experience rapid progression within seven days by >10 folds (OR = 15.72, *p* < 0.001). A multivariable model showed three significant covariates of age, QMD score (≥10%), and IL-6 level (>7 pg/mL) in predicting rapid progression with the best predictive power (AUC = 0.886, 95% CI 0.795–0.974) (Table 4).

### 3.5. Follow-Up Imaging

The mean (±SD) duration between two HRCT images was 9.5 days ± 5.7 (*n* = 82), where the maximum duration was 32 days after the initial scan with available PCR results. Changes in QMD scores were associated with age, the status of rapid progressors, CRP, and QMD at baseline (R^2^ = 0.66, *n* = 82) (Table 5). The overall mean (±SD) reduction was 84% (±0.084) of the QMD score at baseline (*p* < 0.0001), although the rapidly progressing subjects increased the QMD score by 4.9% (*p* = 0.019) compared with the QMD score of the stable (non-rapid progressor) subjects. Older age groups are more likely to increase QMD scores by 0.126, with one unit of increased age in years. At follow-up, CRP was associated with changes in QMD with a mean (±SD) of 0.20 (0.13) (*p* = 0.003). There were 60 subjects within ten days of follow-up CT imaging. These results were mostly consistent with those of the overall subjects. A minor difference was observed in the rapidly progressing subjects, with an increased mean QMD score of 4.6% (*p* = 0.071) compared with the QMD score of the stable subjects. Figure 3 shows a representative longitudinal CT imaging in a rapidly progressive subject and a stable subject.

### 3.6. Longitudinal Changes in Chest CT over Two Months or Longer

Twenty-one patients had longitudinal CT images taken more than two months after the initial diagnosis and hospitalization. CT scans were performed at least twice and up to six times per patient. The final CT scan was performed after discharge. In all patients, lung lesions were still observed with a QTLD of 1.4–29.3% on CT after two months or more. Figure 4a shows the CT image changes in a 58-year-old man. After the COVID-19 diagnosis, a chest CT was performed five times serially from baseline, and the last CT scan was performed 113 days after diagnosis. Chest CT performed at the first hospitalization showed QMD of 12.5%, QGGO of 27%, and QTLD of 39.8%. On the second CT taken after 16 days, the QMD was 22.8%, QGGO was 23.2%, and QTLD was 47%. On the third CT taken after 26 days, the QMD was 14.5%, QGGO was 23.1%, and QTLD was 37.7%. When the patient was discharged, nasal RT-PCR showed an equivocal state, undergoing negative conversion. The fourth CT, after 64 days, showed a QMD score of 1.1% and QGGO of 7.5%. On the fifth CT taken after 113 days, the QMD was 1.4%, QGGO 8.1%, and QTLD 9.7%, mainly GGO lesions remained. The QMD score, CRP level, NLR changes, and RT-PCR results of the other five patients who had longitudinal CT images are shown in Figure 4b. On follow-up CT at three months, the QMD score remained high at 0.5–9.1%, and QGGO remained at 2–19%. In particular, lung lesions remained on CT for several weeks after negative results on real-time RT-PCR assay of SARS-CoV-2 for nasal and pharyngeal swab specimens.

## 4. Discussion

Our results indicated that QCT COVID scores on admission were an independent prognostic factor in predicting rapid progression to severe COVID-19 pneumonia. The quantitative score with the mixed disease pattern (QMD) score was highly predictive, and improve prediction in rapid progression together with other inflammatory markers. Furthermore, as longitudinal trends in CT score were matched with changes in the ratio of neutrophil to lymphocyte and CRP, simple laboratory measurements can be used for monitoring.

Most patients with COVID-19 have mild to moderate symptoms, and their severity has been reduced due to the omicron mutation [35,36]. However, in some patients, pneumonia rapidly progressed within 1–2 weeks after COVID-19 infection, and those patients’ condition worsened rapidly with respiratory failure [37]. Early detection of patients with rapid progression in the early stages of COVID-19 diagnosis is very important for efficient clinical care in a limited medical environment. In a situation where the number of patients is rapidly increasing and the number of hospitalized beds is limited, clinicians can select patients who are expected to progress rapidly, decide on hospitalization, and monitor them, thereby lowering the mortality rate of COVID-19 [38]. The prevalence of severe COVID-19 is approximately 20% [4,5], and the number of patients with rapid progression in this study was 9.6%, which was lower than previously reported. Patient registration in this study was performed when all diagnosed patients were hospitalized without screening in the early stages of the COVID-19 pandemic in South Korea.

Lung CT can provide useful information for diagnosing COVID-19 pneumonia and differentiating it from other diseases [39]. Radiologists interpreted the CT images; however, due to differences in experience and subjective opinion, there was a large variation among radiologists, making it difficult to quantify disease severity as it is both time-consuming and labor-intensive. A previous study has reported the early prediction of disease progression in patients with COVID-19 pneumonia using chest CT and clinical characteristics, wherein various risk factors, including CT severity, CRP, and NLR were analyzed, similar to that in the current study [6]. However, because CT images were reviewed by radiologists and scored according to the degree of involvement, it was difficult to regard the CT severity score as a quantitative indicator. Recently, various studies have reported that the clinical outcome in COVID-19 patients can be predicted using QCT through AI software or deep learning machines [23,40,41,42]. Our study showed that the QCT COVID scores at admission could predict rapid progression in patients with COVID-19. The total lesion volume showed the best performance in assessing COVID-19 pneumonia, which matches the findings of a previous study [41]. The QCT COVID scores are based on radiomic features after deep-learning-based lung and lobar segmentation. Ground glass opacities usually represent acute inflammatory processes, mixed disease is commonly a radiological presentation of interstitial lung disease (ILD), and consolidation is frequently associated with pulmonary infection [33]. In this study, the QMD score had the best predictive power for rapid progression, and the QGGO score remained high until later in long-term follow-up patients.

This study investigated the longitudinal changes in chest CT findings in patients with COVID-19 pneumonia. In approximately 20 patients, lung lesions remained on chest CT 1 month after discharge. It took several weeks or more for the lesions remaining in the lungs to resolve after the RT-PCR assay of SARS-CoV-2 was negatively converted. The follow-up images observed in our study support that symptoms such as cough, sputum, and dyspnea may persist for a long time even after the infection has gone in patients with COVID-19. Even if infectivity disappears after COVID-19 infection, lung lesions can persist for several months, so exposure to other infections should be avoided. Furthermore, if respiratory symptoms persist for a long time, obtaining follow-up images is necessary to check whether the lesion persists or worsens. Conversely, the stable laboratory measurements can be inferred as to stable or resolved COVID-19 disease, which may avoid the unnecessary radiation exposure via CT.

This study had several limitations. First, it was a retrospective study performed on single-center patients to test the utilities of CT imaging and inflammatory markers without developing the prediction model. A larger cohort is needed to validate the utility of QCT COVID scores in assessing the prognosis of patients with COVID-19. Second, this study only included patients from the first half of 2020, so it cannot represent all COVID-19 patients with various coronavirus variants. Additionally, long-term changes could not be observed as follow-up CT images beyond one year were not analyzed. Third, we did not obtain cut-off levels for all variables. Our interest is based on the incremental improvement of AUC as part of evaluation, instead of obtaining a cut-off and obtaining the sensitivity and specificity. As part of the early stages of the work, the main goal of this study was to explore the association and explore the ability of classifying the group of likely rapid progression within a week. Fourth, quantitative scores were used to classify a set of radiomic features which contain high variability in CT technical parameters, such as the kernels and slice thickness. A sharper kernel with a low dose generates grainy noise, and slice thickness leads to volume artifacts and blurs the texture of the images. This study included all CT images with 3-mm slice thickness that reduced the sensitivity and fine characteristics of radiomics, potentially leading to underestimated QCOVID scores [34]. Fifth, we only compared the existing laboratory measurements, but did not compare the existing scoring systems or methods for predicting rapid progression in COVID-19 patients. Lastly, quantitative CT analysis has no internal validation and there may be a risk of overfitting.

## 5. Conclusions

Quantitative CT COVID scores can provide support for a clinician to make informed decisions, in conjugation with laboratory markers, about the rapid progression of COVID-19. The laboratory and monitoring longitudinal changes can be inferred to the change in patients with COVID-19 pneumonia.

## Figures and Tables

**Figure 1 biomedicines-12-00120-f001:**
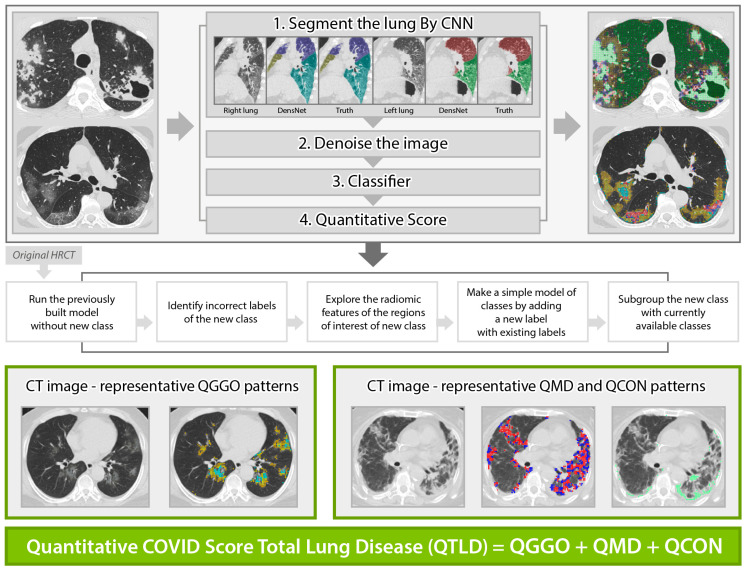
Automated quantitative COVID (QCOVID) score after model adaptation for a new class using five steps: quantitative mixed diseases (QMD: red + blue dots), quantitative ground glass opacity (QGGO: yellow + cyan dots), quantitative consolidation (QCON: mint). Before adaptation, QMD and QGGO classes were available; QCON is a new class.

**Figure 2 biomedicines-12-00120-f002:**
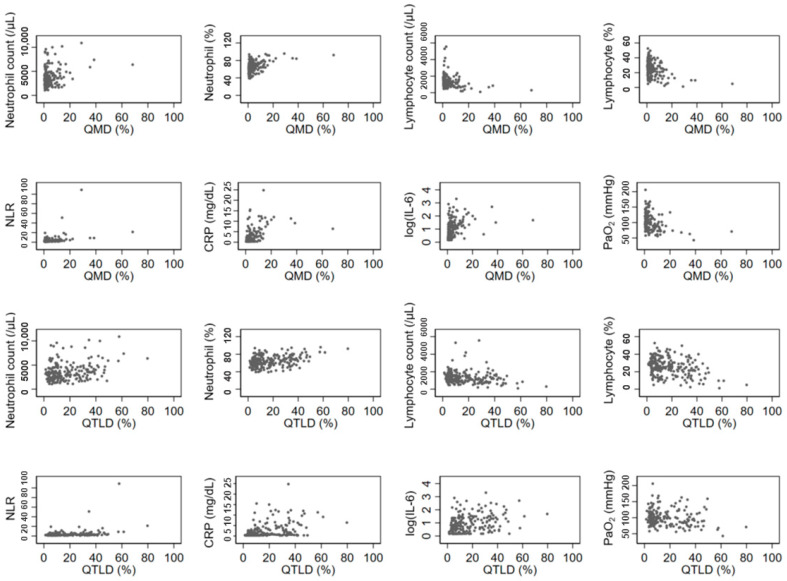
Associations of laboratory and imaging measurement. The QCT COVID scores (QMD and QTLD) are significantly correlated with laboratory measurements of neutrophils, lymphocytes, NLR, CRP, IL-6, and PaO_2_ at baseline.

**Figure 3 biomedicines-12-00120-f003:**
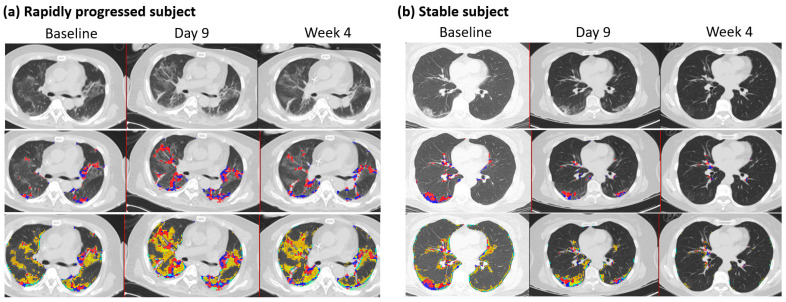
Follow-up CT imaging of rapidly progressed subject and stable subject. (**a**) Rapidly progressed subject: Male, 65 years old, with a BMI of 27. Each column indicates the values at baseline, nine days, and four weeks after HRCT images. Quantitative mixed diseases (QMD) scores are displayed in the second row, with the scores 11.4%, 19.8%, and 12.5% at baseline, nine days, and four weeks in the whole lung. Quantitative total lung diseases (QTLD) are displayed in the third row, with the scores 41.9%, 53.9%, and 43.8% in the whole lung. C-reactive protein scores were 6.3, 5, and 2 at baseline, nine days, and four weeks. The available IL-6 score was 26.8 at week 4. Nasal PCR results were positive at baseline and negative at nine days and four weeks. PCR results by sputum were positive at baseline and equivocal at nine days and four weeks. (**b**) Stable subject: Female, 63 years, with a BMI of 22. Each column indicates the values at baseline, nine days, and eight weeks after HRCT images. Quantitative mixed diseases (QMD) scores are displayed in the second row, with the scores 6.2%, 3.7%, and 0.4% at baseline, nine days, and eight weeks in the whole lung. Quantitative total lung diseases (QTLD) are displayed in the third row, with the scores 19.2%, 11.4%, and 1.6% in the whole lung. C-reactive protein scores were 3.2 and 2.8 at baseline and nine days. IL-6 scores were 18.8 at baseline and 7.9 at nine days. Nasal PCR results were positive at baseline and negative at nine days. PCR results by sputum were positive at baseline and equivocal. Blue and red dots represent the voxels classified as QMD scores (three lungs, last two visits of iDose), yellow dots represent the voxels classified as QGGO scores, and mint dots represent the voxels classified as QCON scores.

**Figure 4 biomedicines-12-00120-f004:**
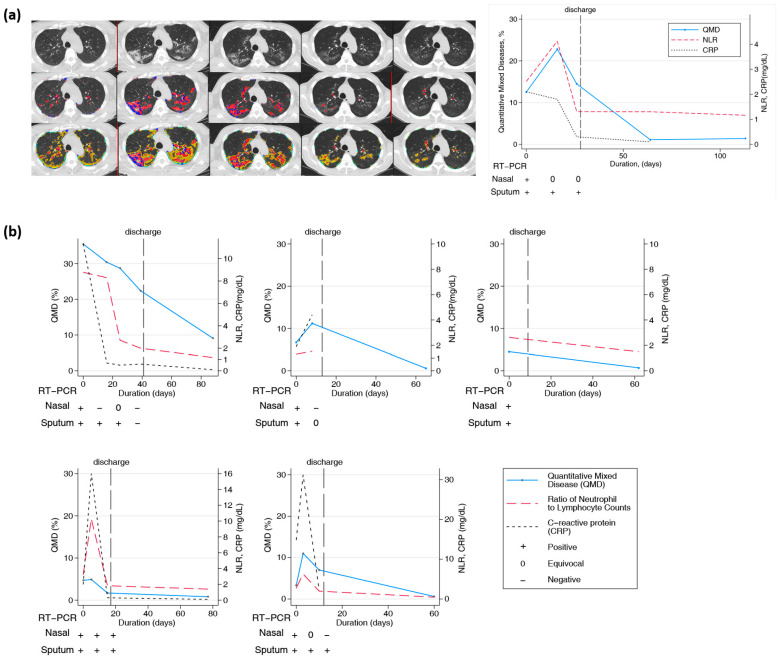
Longitudinal changes in imaging, lab, and PCR measurements of COVID-19 subjects. (**a**) Representative longitudinal CT imaging and CRP, NLR, and PCR results. Blue and red dots represent the voxels classified as QMD score (three lungs, last two visits of iDose), yellow dots represent the voxels classified as QGGO score, and mint dots represent the voxels classified as QCON score. PCR nasal results were positive and equivocal on days 0, 16, and 26. Pharyngeal PCR test results were positive on days 0, 16, and 26. (**b**) Longitudinal changes in quantitative scores by HRCT and laboratory measurements in 5 patients.

**Table 1 biomedicines-12-00120-t001:** Baseline clinical and CT characteristics by stable, rapidly progressive, and overall groups.

Variables	Total (*n* = 218)	Non-Rapid Progressors (*n* = 197)	Rapidly Progressive (*n* = 21)	*p*-Value
*Demographics*, Mean (SD); median (IQR)				
Age	53.33 (16.94) 57 (20)	52.16 (16.92) 55 (20)	64.29 (12.97) 64 (14)	0.0012 ^+^
BMI (kg/m^2^) ^a^	24.26 (3.80) 24.03 (4.18)	24.18 (3.82) 23.9 (4.05)	24.96 (3.56) 25.05 (4.76)	0.2143 ^+^
*Demographics*, *n* (%)				
Male	106 (48.62)	96 (48.73)	10 (47.62)	0.923 *
*Comorbidity*, *n* (%)				
Hypertension	55 (25.23)	48 (24.27)	7 (33.33)	0.428 *
Diabetes Mellitus	40 (18.25)	34 (17.26)	6 (28.57)	0.234 *
Dyslipidemia	17 (7.80)	17 (8.63)	0 (0.00)	0.383 *
Bronchial asthma	3 (1.38)	3 (1.52)	0 (0.00)	>0.999 *
Cancer	11 (5.05)	9 (4.57)	2 (9.52)	0.287 *
Cardiovascular disease	4 (1.83)	4 (2.03)	0 (0.00)	>0.999 *
Cerebrovascular disease	3 (1.38)	2 (1.02)	1 (4.76)	0.263 *
Chronic liver disease	2 (0.92)	2 (1.02)	0 (0.00)	>0.999 *
Chronic kidney disease	2 (0.92)	1 (0.51)	1 (4.76)	0.184 *
Rheumatologic disease	3 (1.38)	2 (1.02)	1 (4.76)	0.263 *
Neurologic disorder	6 (2.75)	5 (2.54)	1 (4.76)	0.459 *
*Clinical Outcome*, *n* (%)				
O_2_ demand	34 (15.60)	13 (6.67)	21 (100)	<0.001 *
Mechanical ventilation	21 (9.63)	0 (0)	21 (100)	<0.001 *
ECMO usage	5 (2.29)	0 (0)	5 (23.81)	<0.001 *
Death	4 (1.83)	0 (0)	4 (19.05)	<0.001 *
*Clinical Outcome*, Mean (SD); median (IQR)				
Duration of stay ^b^ (days)	14.68 (10.92) 12 (11)	13.21 (8.61) 12 (9)	29.83 (18.70) 27.5 (24)	0.0001 ^++^
WBC (/µL)	5420 (2124) 4960 (2380)	5302 (2057) 4900 (2250)	6534 (2454) 6560 (4500)	0.0151 ^++^
PLT (/µL)	201,779 (61,794) 190,500 (95,000)	204,700 (60,780) 193,000 (82,000)	174,430 (66,010) 182,000 (95,000)	0.0517 ^++^
Neutrophil (/µL)	3606 (1840) 3270 (2160)	3445.63 (1686.04) 3200 (1900)	5106.67(2502.19) 4600 (3200)	0.0015 ^++^
Neutrophil, %	64.43 (12.06) 64.4 (16.4)	63.21 (11.45) 62.8 (15)	75.94 (11.74) 73.8 (16.3)	<0.0001 ^++^
Lymphocyte (/µL)	1312.94 (692.4) 1200 (660)	1354.16 (695.5) 1290 (650)	926.19 (537.1) 850 (510)	0.0019 ^++^
Lymphocyte, %	25.60 (10.10) 25.35 (13.7)	26.58 (9.71) 25.8 (12.2)	16.40 (9.13) 18.0 (11.8)	<0.0001 ^++^
Ratio of Neutrophil to Lymphocyte count	4.08 (8.38) 2.56 (2.16)	3.12 (2.56) 2.41 (1.55)	13.11 (24.57) 3.97 (5.55)	<0.0001 ^++^
CRP (mg/dL) ^c^	2.33 (3.48) 0.7 (2.50)	1.77 (2.60) 0.7 (1.80)	7.88 (5.67) 7.0 (7.75)	<0.0001 ^++^
PCT (ng/mL)	0.07 (0.14) 0.05 (0.00)	0.06 (0.11) 0.05 (0.00)	0.17 (0.28) 0.05 (0.04)	<0.0001 ^++^
IL-6 (pg/mL) ^d^	42.13 (163.55) 5.80 (20.5)	37.61 (165.43) 5.45 (17.4)	93.75 (133.79) 43.8 (65.2)	<0.0001 ^++^
*Radiological Outcome*, Mean (SD); median (IQR)				
QGGO CAD, %	12.62 (9.43) 9.3 (14.1)	12.19 (9.34) 9.0 (12.7)	16.67 (9.51) 17.3 (15.6)	0.0321 ^++^
QMD CAD, %	4.30 (6.93) 1.8 (3.7)	3.05 (3.20) 1.6 (2.6)	16.00 (16.16) 12.2 (15.4)	<0.0001 ^++^
QCON CAD, %	0.40 (1.58) 0.10 (0.20)	0.19 (0.34) 0.10 (0.20)	2.33 (4.65) 0.40 (2.9)	0.0002 ^++^
QTLD CAD, %	17.31 (14.05) 12.05 (20.9)	15.43 (11.70) 10.7 (16.1)	35.00 (20.87) 35.7 (21.8)	0.0001 ^++^

^a^: *n* = 185 for non-rapid progressors, *n* = 20 for rapidly progressive; ^b^: *n* = 183 for non-rapid progressors, *n* = 15 for rapidly progressive; ^c^: *n* = 197 for non-rapid progressors, *n* = 20 for rapidly progressive; ^d^: *n* = 194 for non-rapid progressors, *n* = 17 for rapidly progressive. *: *p*-value by Fisher’s exact tests. ^+^: *p*-value by two-sample *t*-test. ^++^: *p*-value by Wilcoxon signed rank-sum test. BMI, body mass index. ECMO, Extracorporeal Membrane Oxygenation. WBC, white blood cell. PLT, platelet. CRP, c-reactive protein. PCT, procalcitonin. IL-6, interleukin-6, CAD, computer-aided design.

**Table 2 biomedicines-12-00120-t002:** Associations between clinical, laboratory, and imaging quantitative COVID-19 scores.

	QGGO, %	QMD, %	QCON, %	QTLD, %	QMD/QTLD, %
	Rho (r)				
	(*p*-Value)				
WBC	0.1553 *	0.1357 *	0.1366 *	0.1857 *	0.0082
	(0.0218)	(0.0453)	(0.0439)	(0.0060)	(0.90)
PLT	−0.0292	−0.1718 *	−0.0570	−0.0721	−0.2208 *
	(0.67)	(0.0110)	(0.4025)	(0.2892)	(0.0010)
Neutrophil Count	0.1990 *	0.2385 *	0.2531 *	0.2533 *	0.0950
	(0.0043)	(0.0004)	(0.0002)	(0.0002)	(0.16)
Neutrophil %	0.1926 *	0.3651 *	0.3728 *	0.2898 *	0.3070 *
	(0.0043)	(<0.0001)	(<0.0001)	(0.0001)	(<0.0001)
Lymphocyte Count	−0.1063	−0.2950 *	−0.2326 *	−0.1889 *	−0.3210 *
	(0.12)	(<0.0001)	(0.0005)	(0.0051)	(<0.0001)
Lymphocyte %	−0.1844 *	−0.3428 *	−0.3209 *	−0.2729 *	−0.2931 *
	(0.0063)	(<0.0001)	(<0.0001)	(<0.0001)	(<0.0001)
NLR	0.1924 *	0.3545 *	0.3405 *	0.2835 *	0.2965 *
	(0.0044)	(<0.0001)	(<0.0001)	(<0.0001)	(<0.0001)
CRP ^a^	0.3270 *	0.5669 *	0.3327 *	0.4298 *	0.4741 *
	(<0.0001)	(<0.0001)	(<0.0001)	(<0.0001)	(<0.0001)
PCT ^b^	0.1757 *	0.2587	0.2090 *	0.2185 *	0.1768 *
	(0.0102)	(0.2697)	(0.0022)	(0.0013)	(0.0097)
IL6 ^c^	0.2560 *	0.4908 *	0.2605 *	0.3389 *	0.4594 *
	(0.0002)	(<0.0001)	(0.0001)	(<0.0001)	(<0.0001)
PaO_2_ ^d^	−0.0801	−0.2547 *	−0.0896	−0.1751 *	−0.3006 *
	(0.2438)	(0.0005)	(0.23)	(0.0184)	(<0.0001)

^a^ *n* = 217 for CRP; ^b^ *n* = 213; ^c^ *n* = 211; ^d^ *n* = 181; * *p* < 0.05.

**Table 3 biomedicines-12-00120-t003:** Prediction of rapid progression using univariate quantitative imaging scores.

Quantitative COVID Score	Odds Ratio (SD)	95% CI Odds Ratio	*p*-Value	AUC	95% CI AUC
QGGO, %	1.05 (0.023)	[1.00, 1.09]	0.043	0.642	[0.513, 0.771]
QMD, %	1.30 (0.068)	[1.18, 1.44]	<0.001	0.813	[0.679, 0.947]
QCON, %	3.71 (1.30)	[1.87, 7.38]	<0.001	0.735	[0.590, 0.881]
QTLD, %	1.09 (0.19)	[1.05, 1.13]	<0.001	0.768	[0.625, 0.910]

**Table 4 biomedicines-12-00120-t004:** Univariate and multivariate analysis of factors to predict rapid progression.

	Univariate-Analysis				Multivariate-Analysis					
	OR (SE)	*p*-Value	95% CI of OR	AUC [95% CI]	OR (SE)	OR (SE)	OR (SE)	OR (SE)	OR (SE)	OR (SE)
Age	1.05 (0.02)	0.002	[1.02, 1.09]	0.716 [0.608, 0.824]					**1.05** **(0.02) ****	1.04 (0.02) ^+^
BMI	1.05 (0.06)	0.38	[0.93, 1.18]	0.585 [0.447, 0.722]						
Male	0.96 (0.44)	0.92	[0.39, 2.35]	0.506 [0.389, 0.625]						
QMD	1.30 (0.07)	<0.001	[1.18, 1.44]	0.813 [0.679, 0.947]						
QMD ≥ 10%	28.31 (15.4)	<0.001	[9.72, 82.3]	0.800 [0.696, 0.905]	15.72 (9.60) **	13.24 (7.50) **		10.21 (6.50) **	**15.94** **(9.85) ****	10.80 (7.02) **
IL-6	1.00 (0.0009)	0.238	[0.992, 1.002]	0.830 [0.746, 0.915]						
IL-6 >7 pg/mL	10.24 (7.85)	0.002	[2.28, 46.0]	0.730 [0.644, 0.816]	4.70 (3.86) *		9.19 (10.35) *	6.57 (7.75) ^+^	**3.36 (2.84) ^+^**	4.55 (5.48) ^+^
CRP	1.40 (0.088)	<0.001	[1.24, 1.58]	0.855 [0.746, 0.964]						
CRP ≥ 1 mg/dL	14.96 (11.37)	<0.001	[3.37, 66.3]	0.762 [0.687, 0.848]		6.09 (4.98) **	4.16 (3.49) **	2.09 (1.93) ^+^		2.00 (1.85) ^+^
AUC [95% CI]					0.864 [0.775, 0.953]	0.856 [0.763, 0.949]	0.802 [0.718, 0.886]	0.868 [0.770, 0.966]	**0.886** **[0.795, 0.974]**	0.882 [0.786, 0.979]

*n* = 205 for BMI, *n* = 211 for IL-6, *n* = 217 for CRP, otherwise *n* = 218; ** if *p* < 0.05, * if 0.05, ^+^ if *p* > 0.10. Bold indicates the highest AUC.

**Table 5 biomedicines-12-00120-t005:** Changes in QMD scores in follow-up CT imaging.

**Changes in Different QMD Scores, *n* = 82, R^2^ = 0.66**				
	**Coefficient**	**SE**	***p*-Value**	**[95% CI]**
PCR Septum positive result	1.437	1.430	0.318	[−1.412, 4.286]
Duration	0.144	0.129	0.266	[−0.112, 0.400]
QMD at baseline	−0.841	0.084	<0.001	[−1.009, −0.673]
Age	0.126	0.050	0.013	[0.027, 0.224]
Rapid Progressors	4.913	2.045	0.019	[0.838, 8.988]
CRP	0.403	0.131	0.003	[0.142, 0.664]
Constant	−4.803	3.245	0.143	[−11.268, 1.662]
**Changes in Different QMD within Ten Days, *n* = 60, R^2^ = 0.44**				
	**Coefficient**	**SE**	***p*-Value**	**[95% CI]**
PCR Septum positive result	−0.178	1.761	0.920	[−3.710, 3.354]
Duration	0.022	0.468	0.963	[−0.917, 0.960]
QMD at baseline	−0.688	0.204	0.001	[−1.097, −0.279]
Age	0.133	0.061	0.034	[0.011, 0.256]
Rapid Progressors	4.598	2.492	0.071	[−0.401, 9.597]
CRP	0.390	0.164	0.021	[0.062, 0.718]
Constant	−3.576	5.429	0.513	[−14.466, 7.313]

## Data Availability

The datasets analyzed during the current study are available from the corresponding author upon reasonable request.

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
