# Peer review of "Quantitative Computed Tomography Lung COVID Scores with Laboratory Markers: Utilization to Predict Rapid Progression and Monitor Longitudinal Changes in Patients with Coronavirus 2019 (COVID-19) Pneumonia"

_biomedicines, 2024, doi:10.3390/biomedicines12010120_

Round 1
Reviewer 1 Report (Previous Reviewer 1)
Comments and Suggestions for Authors
I am afraid that either I have not seen the correct document, or the authors did not address my previous comments. Specifically
Section 2.1, where the material is presented, it would be really useful to illustrate the CT images that will be later analysed, i.e., show a normal CT, one with ground glass opacities, one with reticulation, with architectural distortion and consolidation.
No address neither a rebuttal of why this could not be done.
Figure 1 is a bit cryptic for those not familiar with it. Just to illustrate this, what does the green or brown represent at the top of the image? What is the previously built model?
What are I,j,k,D,x,x^,f,Dnew, etc? A manuscript should stand alone and should not require that a reader has read other papers that have equations which are presumably reproduced in Fig 1. Finally on the figure, I can see the illustrations of QMD QCON etc, but what range of values would these represent? It would be good to show a range of images with different values and show the values of the images.
The authors took the easy route and just cropped out the equations rather than explain them, I still do not understand whtat the brown and green at the top represent or why the images at the bottom do not match the images at the top, or even the 3 images on the bottom right match the 3 on the bottom left.
I will not go on. If authors do not address reviewers' comments, then it is not possible to recommend for publication.
Author Response
We thank the reviewer for providing instructive and important comments, as well as giving us the chance to revise the manuscript. To address the suggestions and concerns of the reviewer, we have modified the manuscript in accordance with the reviewers’ recommendations. Our point-by-point responses are included in a separate file. Thank you for your consideration.

Reviewer 2 Report (New Reviewer)
Comments and Suggestions for Authors
The paper focuses on predicting rapid progression in patients with COVID-19 pneumonia using radiographic scores and laboratory markers, as well as monitoring longitudinal changes. My comments are listed below:
- Provide a clear description of the AI models used for quantitative CT analysis, including details on the training data, architecture, and validation process.
- Include a comparison with existing scoring systems or methods for predicting rapid progression in COVID-19 patients.
- Discuss the clinical implications and potential applications of the findings in more depth.
- Provide more information on the long-term clinical outcomes and prognostic value of the quantitative CT scores.
Author Response
We thank the reviewer for providing instructive and important comments, as well as giving us the chance to revise the manuscript. To address the suggestions and concerns of the reviewer, we have modified the manuscript in accordance with the reviewers’ recommendations. Our point-by-point responses are included in a separate file. Thank you for your consideration.

Reviewer 3 Report (New Reviewer)
Comments and Suggestions for Authors
[General comments]
This manuscript investigates quantitative COVID score computed from CT images. This study is interesting and of clinical value. However, from a technical perspective, there is a lack of the descriptions of the methods. From Figure 1, the quantitative score is calculated following CNN-based lung segmentation, image denoising, and classifier. However, none of the four key steps has been described technically. This is the major failing of the manuscript. Another major concern is about the applicability of the findings on the virus in 2020 to the current virus. Please refer to the specific comments for details.
[Specific comments]
1. From Figure 1, the quantitative score is calculated following CNN-based lung segmentation, image denoising, and classifier. What kind of CNN?
2. How many training sets, validation sets, and test sets for the CNN model?
3. Is there an overfitting issue for the CNN model? Please provide evidence.
4. Do the authors validate their CNN model?
5. What image denoising method is used? What is the purpose of denoising?
6. What classifier? Classifying what?
7. The quantitative score is calculated by QTLD = QGGO + QMD + QCON. How are the three scores in the right side calculated?
8. What do the color pixels in the lung CT images indicate? Do they indicate the scores? If so, please provide a color bar for each sub-image, so the readers can understand that each color represents a score.
9. This is a retrospective study with data collected between February and September 2020. It is known that the virus is constantly mutating. Do the findings on the ‘early versions’ of the COVID apply to the ‘current version’? If not, the value of this study shall be doubtable. If yes, can you provide some evidence?
Comments on the Quality of English LanguageThe English can be improved a little bit.
Author Response
We thank the reviewer for providing instructive and important comments, as well as giving us the chance to revise the manuscript. To address the suggestions and concerns of the reviewer, we have modified the manuscript in accordance with the reviewers’ recommendations. Our point-by-point responses are included in a separate file. Thank you for your consideration.

Reviewer 4 Report (New Reviewer)
Comments and Suggestions for Authors
The topic is interesting and the paper is quite well written. Nevertheless, in my opinion, some parts need to be improved, I have some comments:
1) Abstract. Abstract: Coronavirus disease 2019 (COVID-19), is an ongoing issue in certain populations, present- 19 ing rapidly worsening pneumonia and persistent symptoms. This study aimed to test the predicta- 20 bility of rapid progression using radiographic scores and laboratory markers and present longitu- 21 dinal changes. This retrospective study included 218 COVID-19 pneumonia patients admitted at the 22 Chungnam National University Hospital. Rapid progression was defined as respiratory failure re- 23 quiring mechanical ventilationventilatory support within one week of hospitalization. From high- 24 resolution computed tomography (CT) analyses, quantitative COVID (QCOVID) scores were de- 25 rived: (1) ground glass opacity (QGGO), (2) mixed diseases (QMD), and (3) consolidation (QCON), 26 and the sum, quantitative total lung diseases (QTLD). Laboratory data, including inflammatory 27 markers, were obtained from electronic medical records. Rapid progression was observed in 9.6% 28 of patients. All QCOVID scores predicted rapid progression, with QMD showing the best predicta- 29 bility (AUC=0.813). In multivariate analyses, the QMD score and interleukin(IL)-6 level were im- 30 portant predictors for rapid progression (AUC=0.864). With >2 months follow-up CT, remained 31 lung lesions were observed in 21 subjects, even after several weeks of negative reverse transcription 32 polymerase chain reaction test. AI-driven quantitative scores in conjugation with laboratory mark- 33 ers can be useful in predicting the rapid progression and monitoring of COVID-19. Abstract might be beneficial to include a sentence in the abstract that briefly summarizes the key findings of the study. This can provide readers with a quick overview of the research.
2) Since the COVID-19 pandemic, studies have been conducted to examine the relation- 70 ship between various CT findings and clinical outcomes of patients, and there has been 71 an increased interest in the prognostic value of chest CT at the time of COVID-19 diagno- 72 sis. Various studies have shown that several CT findings (scattered bilateral distribution 73 of lesions, a higher number of involved lobes, the coexistence of diffuse GGO and consol- 74 idations, absence of mixed and reticular patterns, crazy paving, bronchus distortion, etc.) 75 are associated with poor prognosis in patients admitted to the ICU [13-16]. Although the Authors described in detail the findings from the included references, there are several relevant works/reviews, including most important published which should be added and discussed by the Authors:
1- Radiological-pathological signatures of patients with COVID-19-related pneumomediastinum: is there a role for the Sonic hedgehog and Wnt5a pathways?. ERJ Open Res. 2021;7(3):00346-2021. Published 2021 Aug 23. doi:10.1183/23120541.00346-2021
2- Ventilatory associated barotrauma in COVID-19 patients: A multicenter observational case control study (COVI-MIX-study). Pulmonology. 2023;29(6):457-468. doi:10.1016/j.pulmoe.2022.11.002
3-Quantitative CT Metrics Associated with Variability in the Diffusion Capacity of the Lung of Post-COVID-19 Patients with Minimal Residual Lung Lesions. J Imaging. 2023;9(8):150. Published 2023 Jul 26. doi:10.3390/jimaging9080150
4- COVID-19 and Post-Acute COVID-19 Syndrome: From Pathophysiology to Novel Translational Applications. Biomedicines 2022, 10, 47. https://doi.org/10.3390/biomedicines10010047
3) This study explored the predictive value of quantitative CT (QCT) lung COVID 87 scores, along with laboratory inflammation markers including WBC, neutrophil count, 88 lymphocyte count, CRP, procalcitonin, and IL-6, for rapid progression within one week 89 of hospitalization. Additionally, we investigated the changes and resolution of lung le- 90 sions in patients with COVID-19 pneumonia based on longitudinal CT imaging data. Please improve the description of this part and underline the novelty of the study.
4) Figures 1 - 4. If the images can be higher resolution that would be better for the reader.
5) 3. Results 181 3.1. Patient characteristics 182 A total of 225 patients with COVID-19 pneumonia were identified according to the 183 inclusion criteria, and seven were excluded for the following reasons: (1) CT scores were 184 not available due to slice thickness >3 mm or no available DICOM images (n=5); (2) dif- 185 ferent baseline scan date (n=2). Please, underline the most important results to clarify the data.
6) 4. Discussion 309 Our results indicated that QCT COVID scores were associated with inflammatory 310 markers and that QCT COVID scores on admission were independent predictors of rapid 311 progression to severe COVID-19 pneumonia. The QMD score had high predictive power, 312 and rapid progression could be better predicted if the QMD score and other inflammatory 313 markers were used together...... The discussion section needs to be improved. It could be interesting to record the aim of the study. It is necessary to be more concise in the presentation of the facts, clarifying the results obtained and comparing them with previous or similar studies. However, it is interesting to answer the questions that arise from these results, backed up by published literature.
7) 5. Conclusions 380 Quantitative CT analysis using QCT COVID scores, in conjugation with laboratory 381 markers, may be an effective and important method for assessing the rapid progression 382 of COVID-19 and monitoring longitudinal changes in patients with COVID-19 pneumo- 383 nia. Please, improve the conclusions. I suggest to underline the novelty of the study and the possible clinical implications.
Comments on the Quality of English LanguageMinor changes of English language are required
Author Response
We thank the reviewer for providing instructive and important comments, as well as giving us the chance to revise the manuscript. To address the suggestions and concerns of the reviewer, we have modified the manuscript in accordance with the reviewers’ recommendations. Our point-by-point responses are included in a separate file. Thank you for your consideration.

Round 2
Reviewer 1 Report (Previous Reviewer 1)
Comments and Suggestions for Authors
Authors have addressed my previous concerns
Reviewer 2 Report (New Reviewer)
Comments and Suggestions for Authors
The authors addressed all my previous comments
Reviewer 3 Report (New Reviewer)
Comments and Suggestions for Authors
Thanks for the revision. It has addressed my concerns.
Comments on the Quality of English LanguageMinor editing is needed.
Reviewer 4 Report (New Reviewer)
Comments and Suggestions for Authors
The manuscript has been improved as requested, I have no further comments
Comments on the Quality of English LanguageMinor changes of English language are required
This manuscript is a resubmission of an earlier submission. The following is a list of the peer review reports and author responses from that submission.
Round 1
Reviewer 1 Report
Comments and Suggestions for Authors
The manuscript submitted by Da Hyun Kang and co-authors describes a methodology to assess the condition of patients who present Covid in clinical settings in order to predict if them are likely to progress to intensive care and further care. The methodology is based on imaging, specifically Computed Tomography images and merging other markers derived from blood tests.
This work is interesting, has clinical applicability and is inherently interdisciplinary. It provides good results showing the power of the methodology. The presentation could be improved before it can be recommended for publication and the comments below will highlight some areas that need to be improved as at it stands, the manuscript could be difficult to reproduce and there are many assumptions that require educated guesses from a reader.
Specifically,
L41, ICU has not been defined, I can guess it refers to intensive care units, but I should not be assuming the meaning of acronyms, these should always be defined the first time they appear in a manuscript.
L55 LDH has not been defined
L61 / L74 Covid is sometimes written COVID sometimes Covid, there should be consistency along the manuscript.
Section 2.1, where the material is presented, it would be really useful to illustrate the CT images that will be later analysed, i.e., show a normal CT, one with ground glass opacities, one with reticulation, with architectural distortion and consolidation. Think of a younger reader who would greatly benefit from identifying these important features.
Section 2.3 is very important and as it stands it is not easy to understand or to follow. Even if based on published material, a short description of the pipeline (L112) should be included. What domain adaption was applied and how?
L122, How exactly are the scores calculated?
Figure 1 is a bit cryptic for those not familiar with it. Just to illustrate this, what does the green or brown represent at the top of the image? What is the previously built model?
What are I,j,k,D,x,x^,f,Dnew, etc? A manuscript should stand alone and should not require that a reader has read other papers that have equations which are presumably reproduced in Fig 1. Finally on the figure, I can see the illustrations of QMD QCON etc, but what range of values would these represent? It would be good to show a range of images with different values and show the values of the images.
In section 2.4, statistical analysis, please specify how the tests were performed, specifically, there are p-values shown in the tables but to me it was not clear exactly how these were done. This is important because in Table 1, Age shows a p-value of 0.0012. This suggest that the rapid progressive and the non-rapid progressors can be perfectly distinguished by age. If this is correct, then it could imply that there is no need even to expose a patient to the radiation of a CT scan or any blood tests, just ask age 35? Will be fine, 75, will end in ICU.
Figure 2 shows interesting scatterplots, but there are a few outliers, the authors should make a quick comment of what these outliers represent.
Comments on the Quality of English Language
English seems fine to me.
Author Response
We thank the reviewer for providing instructive and important comments, as well as giving us the chance to revise the manuscript. To address the suggestions and concerns of the reviewer, we have modified the manuscript in accordance with the reviewers’ recommendations. Our point-by-point responses are included in the attached file. Please see the attachment.

Reviewer 2 Report
Comments and Suggestions for Authors
Please avoid using abbreviations in the title of the manuscript.
Abstract
- In the title is about CT score but the aim states that "radiographic scores and laboratory markers at baseline and present longitudinal changes".
- The abstract is not sufficiently specific and lacks clarity.
- Please closely follow the requested structure.
- Results should be specific.
Define abbreviations in the keywords.
Introduction
- "Recently," did not fit with the reality of 2023.
- Define ICU abbreviation.
- "The prevalence of severe COVID-19 is 15.7-26.1% in hospitalized patients" where?
- "abnormal laboratory and chest computed tomography (CT) findings" please list them with measurements of occurance.
- "several studies" several does not fit with two references.
- "has been reported differently in each study" provide the ranges of these cut-off values (min to max with linked references).
- It is unclear which markers had been already evaluated and thus which is the added values of your study.
- It is unclear how the authors expected to have changes in QCT in one mount to support their aim.
Methods
- Do not start a sentence with an abbreviation (e.g., line 87, 172, 214, 308, 338 etc.).
- Which variant of COVID-19 was in place at the time of the study?
- Please provide the date of ethics approval.
- Define CRP, and IL abbreviations.
- "2.3. Quantitative high-resolution computed tomography (HRCT) imaging analyses" should be read as "2.3. Quantitative high-resolution computed tomography imaging analyses"
- Define HRCT abbreviation in line 102.
- "study population" should be read as "study cohort"
- "(JGG and FA)" this is unclear.
- "radiological patterns" on CTs?
- The methods used to classify lung CT changes are not described in sufficient detail to allow the reproduction of the study. Did you use specific software for classification? If yea, who validated the automated score?
- The used score is not described sufficiently to allow reproduction.
- "O2 demand" a subscript is missing.
- Define "Non-rapid progressors" and "Rapidly progressive"
Results
- It is unclear the severity of the patients included in the cohort.
- The "N" in Table 1 should be read as "n". This comment apply along the manuscript.
- Please delete "%" from the body of tables.
- It is unclear which tests were used in comparisons.
- "1.00" should be read as ">0.999"
- At a sample of 21 patients the normal distribution is not appropriate to be tested with statistical tests, so the parametric tests are not appropriate.
- I doubt that when a group is with 0%, a test would be appropriate (violation of tests assumption).
- Please do not duplicate results in text and tables.
- "3.3. Association between quantitative CT (QCT) lung COVID score and laboratory findings" should be read as "3.3. Association between quantitative CT lung COVID score and laboratory findings"
- I am not convinced that CT scores as a qualitative variable (the percentages of the presence of a specific pattern) are appropriate for the correlation analysis. Furthermore, it is unclear which correlation coefficient was applied. It would be useful to have the scatter plots associated with these correlations (r generally stands for Pearson correlation coefficient). According to the scatter plots, you gave some outliers - after removal no such significances will be reliable.
- Please provide the associated 95% confidence intervals associated to AUC. Also, provide the cut-off values in the table.
Discussion
- 272-273 this is duplicated information; please delete it.
- I could not find any reference to your own tables and figure.
- "Our study demonstrated that the QCT" this statement is too strong considering the number of evaluated patients.
Conclusion
- "monitoring COVID-19" it should be read as "progression"?
Author Response

(The authors gave the same response as above.)

Round 2
Reviewer 2 Report
Comments and Suggestions for Authors
Your manuscript looks better, but not sufficient.
- Rapid progression with changes in CT score is not feasible because the scores would not significantly modify quickly. I would recommend using the term "disease progression" and having specific objectives related to rapid progression.
- In the methods section, you present the follow-up on HRCT but I was not able to find this information in the results section. What were the scores in the follow-up?
- Prediction means that you test the model on data that were not included in the development of the model. It is prediction or estimation mode?
- If you want to to estimate the disease progression, you must include in the modele only the baseline measurements. It is not clear in your manuscript which data were included in the models.
- You changed in the methods section parametric test with non-parametric tests but the p-values remain the same; the change to be true is almost null.
- "analyses were used for evaluation" evaluation of what?
- It is unclear which was the outcome variable in the multivariable linear regression and if or not the assumptions were verified. It is unclear how independent variables were chosen to enter the multivariable linear regression.
- The methods used to evaluate "longitudinal changes" are not presented in details.
- It is still unclear which tests were used in Table 1. For example 3 vs. 0 patients will not fit to any test assumption, but you reported a p-value.
- In table 1 you have data that according to mean and standard deviation does not follow the normal distribution (e.g., 16.00 (16.16)) so these summary statistics are not appropriate as neither is the comparison with a parametric test.
- In the results section, please do not write how the graph could be interpreted.
- It is still unclear how AUC reported in Table 4 was obtained.
- In table 5
* if QMD score is the outcome variable it is not correct to have QMD baseline as an independent variable because it is, in fact a dependent variable.
* it is unclear how Changes in different QMD scores were calculated.
* it is unclear which statistical method was applied here.
Author Response
We thank the reviewer for providing instructive and important comments, as well as giving us the chance to revise the manuscript again. To address the suggestions and concerns of the reviewer, we have modified the manuscript in accordance with the reviewers’ recommendations. Our point-by-point responses are included in the attached file. Please see the attachment.

Round 3
Reviewer 2 Report
Comments and Suggestions for Authors
Your paper looks better but still needs improvements before publication.
Comments
- Please provide the 95% confidence interval for AUCs to ensure an appropriate interpretation.
- "has been reported differently in each study" from this result that different methods were used to identify the threshold.
- "upon the normality assumption" the work assumption is not appropriate.
- The presented statistical method still lacks details to allow replication/reproduction.
- I do not agree with your response to "Point #10." Ref [4] indicated in your answer does not support your explanation. Please read one of the paper of that author in ref [4] 10.1213/ANE.0000000000002471
- It is still unclear the tests applied in Table 1 for each variable.
- The *s in table 2 are redundant.
- You reported Spearman's correlation coefficient but included the line that show a linear relation in Figure 2. Do not include the line in Figure 2.
- The 95% confidence intervals associated to AUC are still missing along the manuscript.
- When you report the AUC you must also report the cut-off value (in the results section) along with the method used to identify (in the Methods section; Youden?).
- "(1.02, 1.09)" round brackets tells the reader that the OR associated to age is expected to be in the population higher than 1.02 and smaller than 1.09 (the value not included in the range). Is this true?
Author Response

(The authors gave the same response as above.)

Round 4
Reviewer 2 Report
Comments and Suggestions for Authors
Thank you for considering all my suggestions. Your manuscript looks better; congrats.